# Natural Deep Eutectic Solvent-Assisted Extraction, Structural Characterization, and Immunomodulatory Activity of Polysaccharides from *Paecilomyces hepiali*

**DOI:** 10.3390/molecules27228020

**Published:** 2022-11-18

**Authors:** Yanbin Wang, Feijia Xu, Junwen Cheng, Xueqian Wu, Juan Xu, Chunru Li, Weiqi Li, Na Xie, Yuqin Wang, Liang He

**Affiliations:** 1The Key Laboratory of Biochemical Utilization of Zhejiang Province, Zhejiang Academy of Forestry, Hangzhou 310023, China; 2Zhejiang Provincial Key Laboratory of Resources Protection and Innovation of Traditional Chinese Medicine, Zhejiang A&F University, Hangzhou 311300, China; 3Bioasia Life Science Institute, Zhejiang Bioasia Pharmaceutical Co., Ltd., Pinghu 314200, China; 4College of Life Sciences, Zhejiang University, Hangzhou 310058, China

**Keywords:** deep eutectic solvent, *Paecilomyces hepiali*, polysaccharides, physicochemical property, immunomodulatory activity

## Abstract

Polysaccharides, which can be affected by different preparations, play a crucial role in the biological function of *Paecilomyces hepiali* (PHPS) as a health food. To explore high-valued polysaccharides and reduce the negative influence of human involvement, a green tailorable deep eutectic solvent (DES) was applied to optimize the extraction of polysaccharides (PHPS-D), followed by the evaluation of the structural properties and immunomodulation by comparison with the hot-water method (PHPS-W). The results indicated that the best system for PHPS-D was a type of carboxylic acid-based DES consisting of choline chloride and succinic acid in the molar ratio of 1:3, with a 30% water content. The optimal condition was as follows: liquid–solid ratio of 50 mL/g, extraction temperature of 85 °C, and extraction time of 1.7 h. The actual PHPS-D yield was 12.78 ± 0.17%, which was obviously higher than that of PHPS-W. The structural characteristics suggested that PHPS-D contained more uronic acid (22.34 ± 1.38%) and glucose (40.3 ± 0.5%), with a higher molecular weight (3.26 × 10^5^ g/mol) and longer radius of gyration (78.2 ± 3.6 nm), as well as extended chain conformation, compared with PHPS-W, and these results were confirmed by AFM and SEM. Immunomodulatory assays suggested that PHPS-D showed better performance than PHPS-W regarding pinocytic activity and the secretion of NO and pro-inflammatory cytokines (IL-6, TNF-α and IL-1β) by activating the corresponding mRNA expression in RAW264.7 cells. This study showed that carboxylic acid-based DES could be a promising tailorable green system for acidic polysaccharide preparation and the valorization of *P. hepiali* in functional foods.

## 1. Introduction

Cordyceps, commonly known as chongcao, is a desiccation stroma and insect body parasitized by symbiotic fungi on larvae of Batmanidae, Lepidoptera, which can be found in the southern part of China at an altitude of 3500–5000 m [1]. Compared with other famous traditional Chinese herbal medicines, the severe growth environment and limited collection of this variety lead to an extreme scarcity and high value of this natural resource. Therefore, significant studies have been focused on the field of fungi isolation, morphology, man-made cultivation, and fermentation [2]. *P. hepiali*, as one of the edible and medicinal fungi from the fruiting body of natural Cordyceps, has been widely researched recently due to its richness in bioactive compounds, including polysaccharides, flavonoids, amino acids, organic acids, nucleosides, and steroids [3,4]. Among them, polysaccharides have been reported to present antioxidant, anti-hypertension, anti-inflammatory, anticancer, and other biological activities [5,6,7,8]. Those biological features are closely correlated with the structural information of polysaccharides, including molecular weight, monosaccharide components, flexible branches, and glycosidic linkages [6]. 

Many studies have showed that the structural properties of polysaccharide can be affected by different preparations [7]. The current extraction strategies mainly focus on conventional processes (hot water extraction, alkali-alcohol extraction, microwave extraction, subcritical water extraction, etc.), which not only have inherent limitations, such as long extraction time, high energy consumption, and low extraction efficiency, but also show poor selectivity [9,10,11]. Some widely used ionic liquids are still unsatisfactory because of their high cost, poor biocompatibility, and adverse recovery. In consideration of the aforementioned drawbacks, a new type of deep eutectic solvent system is drawing great attention for its potentially commercial application in the health food industry [12].

Deep eutectic solvents (DESs) refer to a two-component or three-component eutectic mixture composed of hydrogen bond receptors (HBA), such as quaternary ammonium salt, choline chloride, betaine, etc., and hydrogen bond donors (HBD), such as amide, carboxylic acid, polyol, etc., according to a certain stoichiometric ratio, which can cause the dissolution of a variety of substances by the acceptance of protons or electrons from the external environment [13]. Based on the large asymmetric ions of HBA and low lattice energy of HBD, the character of their conformational flexibility endows it with tailorable physicochemical properties, which can affect reaction selectivity, avoid biopolymer chain scission, and highly dissolve the target [14]. Such properties give them striking qualities, including higher safety, better biocompatibility, more excellent biodegradability, lower solvents consumption, more distinctive reaction, and higher selective solubility compared with other extraction methods [14,15]. Therefore, the successful completion of polysaccharide extraction by preparing different types of DESs have been reported in various plants, such as *Indocalamus tessellatus* [16], sweet tea [17], and *Averrhoa bilimbi* [18]. All the studies have shown that DESs were conducive to the system, with high solubility, selectivity, non-volatility, thermostability, and eco-friendship [12,13]. Unfortunately, as yet, little research has been done in the field of polysaccharide extraction from *P. hepiali* using natural deep eutectic solvent assistance.

Hence, the goal of this research is to seek a green and tailorable way to extract specific polysaccharide from cultured mycelia for valorization of *P. hepiali*. In this study, different types of DES formula were originally screened for the extraction process with ultrasonic assistance. Then, the single factors (ultrasonic power, water content, extraction temperature) of the experiment were combined with response surface methodology to optimize the extraction conditions of polysaccharides from *P. hepiali* (PHPS). Moreover, the comparison of the primary physicochemical properties of polysaccharides prepared by the DES method (PHPS-D) and hot-water extracted polysaccharides (PHPS-W) were analyzed by Fourier infrared chromatography, molecular weight, and monosaccharide composition, which was confirmed by the morphological analysis of cell walls under different treatments using scanning electron microscopy. Finally, the immunomodulatory effects of two different prepared polysaccharides on cytokine release were studied, based on murine macrophage RAW264.7 cell culture for a better understanding the advantages of the DES system. 

## 2. Results

### 2.1. Screening of DESs System for PHPS Extraction

#### 2.1.1. The Optimal DESs System

The investigation of the optimal DES for polysaccharide in a solid–liquid system is the key step during the extraction process. Different types of DESs have different physicochemical properties, including viscosity, solubility, and polarity, which significantly affect the yield and extraction efficiency [12]. In this study, choline chloride and betaine were used as HBAs, and 1,2-propylene glycol, malonic acid, DL-malic acid, glycerol, D-sorbitol, citric acid, succinic acid, urea, 1,4-butanediol, ethylene glycol, 1,3-butanediol were used as the HBDs to constitute different DESs. During the preparation, it was regrettable that some DESs (DES-8, DES-9, DES-11, DES-12, DES-13, DES-14, DES-15) could not form a transparent solution by recrystallization, or were leftover at the bottom. This phenomenon might occur when the inappropriate size of the cations and their conformation flexibility in some types of DESs cause large lattice enthalpies and small entropy changes [15]. Therefore, it was crucial to find the proper components of HBA and HBD for DES.

The results for the remaining eight groups of DESs (DES-1, DES-2, DES-3, DES-4, DES-5, DES-6, DES-7, DES-10) are shown in Figure 1A. It shows that the range of PHPD-D yield extracted by these eight DESs was 6.23–11.56%, and the system composed of Chcl-MA was more suitable for the PHPS-D extraction compared with the others. This may be due to the fact that the Chcl-SA possesses the proper viscosity and polarity for the process, which reduced the resistance of the target molecule of the cell surface broken down by the hydrogen bonds. Moreover, there was a higher binding affinity between Chcl-SA and PHPS-D. It seemed that the carboxylic acid-based DES was favorable to extract the polysaccharides of *P. hepiali* compared to other different HBD. Therefore, Chcl-SA was selected as the optimal extraction solvent for use in subsequent experiments. 

#### 2.1.2. Screening of Optimal Molar Ratio and Water Content

The molar ratio of HBA:HBD may significantly affect the chemical and physical characteristics, including the viscosity and surface tension [13]. Figure 1B shows that there was a steady increase of the extraction yield with the increasing molar ratios, and the highest extraction yield was obtained when the molar ratio of choline chloride to succinic acid was 1:3. The yield of PHPS-D climbed to 12.21%. The proper viscosity and better surface tension at this level may be attributed to that phenomenon. This kind of solvent may have a stronger ability to form hydrogen bonds by accepting protons or electrons in the external environment, which would show good performance in dissolving various substances and promoting the permeability of polysaccharide from the cell wall. Hence, the mole ratio of 1:3 in Chcl-SA was considered as the best formula to carry out the extraction of PHPS-D. 

It is well known that water content can directly change the polarity and viscosity of a solvent, thus improving the mass transfer rate of the compounds [12,15]. It can be found that the extraction yield of PHPS-D progressively increased at the initial stage until a peak is reached at a 30% water content, at which time it dramatically decreased to 9.28%, as shown in Figure 1C. The maximum production of PHPS-D reached the value of 12.23% at the top level, which was three-times higher than that reached at 50% water content. The reason could be ascribed to the weakness of the existed hydrogen bonds of Chcl-SA and the faster movement of PHPS-D from the inner parts of the cell matrices. Nevertheless, this advantage would be eliminated by the augment of polarity and the risk of the insolubility of PHPS-D when the water content is over 30%. Based on the above results, 30% was adopted as the suitable water content for the constituents of Chcl-SA.

### 2.2. Single Factor Evaluation

#### 2.2.1. Optimal Extraction Time

Extraction time is usually one of the vital factors that influence the yield of polysaccharide. Six different levels of time were tested to yield the production of PHPS-D, as shown in Figure 2A. The profile illustrates that at the early stage of extraction, intracellular polysaccharides had low diffusion resistance due to the destruction of the cell structure by DES, which improved the initial extraction yield remarkably with the extended extraction time up to 1.5 h (*p* < 0.05). Afterward, the value of yield did not increase, but tended to slightly decline. This may be because under the same condition, an extraction treatment that is too long may lead to the structural damage and degradation of PHPS-D, to some extent. Hence, the extraction times for the polysaccharide preparation were set to be 1.0–2.0 h.

#### 2.2.2. Optimal Liquid–Solid Ratio

Considering its significant impact on the extraction efficiency of polysaccharides, the effects of various liquid–solid ratios from 10:1 mL/g to 60:1 mL/g on the extraction were studied, as shown in Figure 2B. Similarly, the lower percentage of the solvent was able to dissolve the relatively saturated polysaccharides during the initial process below 40:1 mL/g, in which PHPS-D molecules could be steadily transferred into the solution, resulting in the increase in the extraction yield up to 12.43%. Thereafter, with the addition of the ratio of solvent to material, the PHPS-D yield began to decrease moderately, which suggested that the enhancement of the liquid–solid ratio had little effect on the extraction of PHPS. In order to pursue the highest extraction efficiency and the most efficient cost of follow-up treatment, the range of 30:1–50:1 was selected for subsequent experiments.

#### 2.2.3. Optimal Extraction Temperature

Temperature has been commonly believed to furnish energy and force to speed up the movement of the target molecules for the system. In this study, the effect of six extraction temperatures on the PHPS-D content was investigated, and the results are presented in Figure 2C. As expected, the curve of PHPS-D yield tended to grow steadily within the tested range, which meant that the extraction yield progressively increased with the augmentation of the temperature, reaching an optimized value of 12.51% when the temperature reached 100 °C. This revealed that increasing the temperature promoted the rate of molecular diffusion with the mass transfer of the target compound [18]. Different from the findings for traditional hot-water extraction, there was no obvious degradation of the polysaccharide at a relatively high temperature. With respect to the thermal susceptibility of biomacromolecules and energy saving, the temperatures ranging from 80–100 °C were selected as the examined parameters for the following experiments.

### 2.3. Optimization by RSM

#### 2.3.1. BBD Analysis

On the basis of the analysis of the single factor assay, the response surface method was conducted to carry out the experimental design of three factors and three levels, including 17 groups of experimental schemes, and each group was tested three times in parallel (Table 1). For the investigation of the optimized condition, a regression model of prediction was established by the following quadratic regression equation using the help of DesignExpert software: y = 12.47 + 0.59 X_1_ − 0.056 X_2_ + 0.19 X_3_ − 0.20 X_1_X_2_ − 0.11 X_1_X_3_ − 0.38 X_2_X_3_ − 0.31 X_1_^2^ − 0.45 X_2_^2^ − 0.30 X_3_^2^(1)

Table 2 showed the variance analysis of each variable on the extraction process of PHPS-D, as well as the validity of the fitted mathematical model. The *p*-value (less than 0.0001) and *F* value (52.63) confirmed the significance of the regression model, and the *p*-value of the lack of fit (0.0836 > 0.05) and *F*-value (4.73) further validated that the predicted model was sufficient to accurately represent the tested data [11]. Moreover, the coefficient of determination (R^2^) was 0.9854, which reflected that 98.54% of the variables could be determined by the model. A total of 0.9667 of the adjusted coefficient of determination (R^2^_adj_) correlated well with the R^2^, which suggested that the model can explain 96.67% of the changes in response value. The lower coefficient variation (C.V.% = 0.92) proved that the regression model was reliable and accurate. Further, the *p* values of X_1_, X_3_, X_1_X_2_, X_2_X_3,_ X_1_^2^, X_2_^2^, and X_3_^2^ were all less than 0.01, showing a level of extreme significance for all the variables, except for X_2_ and X_2_X_3_ (*p* > 0.05). The effects of the three parameters on the PHPS-D yield was as follows: liquid–solid ratio > extraction time > extraction temperature.

#### 2.3.2. Interactive Effects on PHPS-D Yield

Figure 3 visually shows the 3D and contour map of the DES-assisted extraction regarding PHPS-D yield. It can be noticed that the steep trends in the 3D plots of X_1_X_2_ and X_1_X_3_ and the elliptical contour shapes in Figure 3A,C reflect the significant effects of two interaction factors on PHPS-D yield, which were liquid–solid ratio and extraction temperature, and liquid–solid ratio and extraction time, respectively. Moreover, the liquid–solid ratio had a more positive influence on the extraction than the other two factors. When a higher value of the liquid–solid ratio was adopted in the process, larger concentration gradient could be achieved during the mass transportation, resulting in an enhancement of PHPS-D yield. However, when the value was over 45 mL/g, the system was not ample to obtain the ideal result, suggesting that the proper Chcl-SA environment is essential for the acquisition of PHPS-D. Thus, it is requisite to find a precise liquid–solid ratio to obtain the optimal driving force for the extraction. The circle contour plot and relative mild curve shown in Figure 3B prove that the mutual interaction of extraction temperature and time had no significant effect on the PHPS-D yield within the tested range (*p* < 0.0753) [16]. Considering the above results, it was convincing that the influence of the three parameters was consistent with the prediction of the regression model and ANOVA analysis, and DES may play an important role during the extraction process.

### 2.4. Validation

The optimal extraction process of *P. hepiali* polysaccharides was analyzed as follows: the liquid–solid ratio was 50.35 mL/g, the extraction temperature was 85.30 °C, the extraction time was 1.71 h, and the PHPS extraction rate was 12.83%. Considering the real operability, the predicted process conditions were modified as: liquid–solid ratio of 50 mL/g, extraction temperature of 85 °C, and extraction time of 1.7 h. Under this condition, three parallel experiments were verified to obtain an average rate of 12.78 ± 0.17%, which was slightly lower than the predicted value, with a 0.39% error. 

### 2.5. Physicochemical Properties of PHPSs

#### 2.5.1. Yield and Chemical Composition

Starting from the raw mycelia of *P. hepiali*, we were pleasantly surprised that the extraction yield of PHPS by the optimized DES system (Chcl-SA) reached 12.78 ± 0.17% in Table 3, while the PHPS obtained by the traditional hot water extraction method was only 8.67 ± 0.21%, (9.37% in the previous study) [6,10]. Considering the extracted yield, it was inspiring to discover that the developed DES extraction could promote the release of PHPS from the inner cell wall of *P. hepiali*. Moreover, the total sugars in PHPS-D and PHPS-W were 80.92 ± 1.23% and 74.53 ± 2.36%, with minor amounts of protein inside. In addition, the uronic acid of PHPS-D was 22.34 ± 1.38%, which was nearly five times higher than that of PHPS-W (5.28 ± 0.16%). During the process, the Chcl-SA system could favorably promote the mass transfer by hydrolyzing the bonds between the *P. hepiali* matrices and the polysaccharides, which would be more beneficial for the release of uronic acid from the cell wall. The higher yield of PHPS-D further revealed that the acidic character of Chcl-carboxylic acid DES exhibited a strong solubility on acidic polymers from raw materials. A similar phenomenon has been found in the preparation of acidic polysaccharides from lotus leaves [19].

#### 2.5.2. Molecular Weights of PHPS

As many studies have shown, the molecular weight is one of the intrinsic properties of biopolymers showing their biological activity, which could be differently extracted by various methods [20]. Figure 4 illustrates the molecular weight profiles of PHPS-D and PHPS-W. It was found that PHPS-D was eluted from the SEC column as a major peak of 40–45 min (98.2% mass fraction) and a small peak of 50–60 min (1.8% mass fraction), while PHPS-W was separated into two fractions, with a majority at 50–60 min (93.1% mass fraction) and small signal at 40–45 min (6.9% mass fraction). It also can be found that the red line reflecting the molar mass of every eluted PHPS-D particle was in the range of 1.0 × 10^5^ to 1.0 × 10^6^ g/mol at the whole elution process. The blue line indicates that the molar mass value of the PHPS-W particle drops down into the range of 1.0 × 10^4^ to 1.0 × 10^5^ g/mol from 45 min to 60 min, although it is similar with that of PHPS-D before 45 min. Both of the eluted curves reflected the near homogeneity of molecular size distribution using two different preparations. After the determination by the MALLS system, the Mw and Rg of polysaccharides in PHPS-D were 3.26 × 10^5^ g/mol and 78.2 ± 3.6 nm, while the values for PHPS-W were 4.37 × 10^4^ g/mol and 20.1 ± 2.7 nm (Table 3), respectively. Interestingly, the Mw of PHPS-W prepared by the traditional hot-water extraction method was smaller than that of PHPS-D. In this respect, the single molecular chain of the polysaccharide treated by carboxylic acid DES tended to be more flexible, and extended it architecture more significantly in the aqueous method than in the conventional one. The reason could be that there was a stronger interforce and hydrogen bonding formation between Chcl-SA and the intracellular polysaccharides of *P. hepiali*, which would be attributed to the penetration of active targets from the inner cell walls of the raw mycelia to the surface [21].

#### 2.5.3. Monosaccharide Compositions of PHPS

The type of monosaccharides inside the polysaccharide is believed to greatly affect its bioactivity due to the intrinsic bioinformation for special recognition of the target [21,22]. Figure 5 presented the HPLC eluting curves of 10 mixed standard monosaccharides and two different samples after PMP derivatization. The data suggested that there was a significant difference in molar ratios between PHPS-D and PHPS-W, although they had similar sugar compositions. Glucuronic acid (22.5 ± 0.4%) and glucose (40.3 ± 0.5%) were the major components of PHPS-D, indicating that it might exist as one type of glucan, while glucose (30.1 ± 0.4%), galactose (20.6 ± 0.5%), and arabinose (22.4 ± 0.7%) were the dominant sugars in PHPS-W. Furthermore, the amount of GlcUA in PHPS-D was over 4-fold higher than that in PHPS-W (Table 3), with a relative content of 22.5 ± 0.4%. These results were in a good agreement with the analysis of chemical properties. The reason could be that the hydrogen bond in DES showed better performance regarding the extraction of the uronic functional groups in the biopolymers from the raw mycelia of *P. hepiali*. The findings confirmed that the tailorable DES may specifically facilitate the digestion of acidic polysaccharides from the cell wall of the natural materials [19].

#### 2.5.4. FT-IR Analysis

Both PHPS-D and PHPS-W presented the characteristic properties of polysaccharides, with a slight difference, shown in Figure 6. The prominent band at 3396 cm^−1^ was due to the hydroxyl stretching vibration of the polysaccharide. The peak at 2935 cm^−1^ was attributed to the C-H stretching vibration. The strong absorption band at 1640 cm^−1^ was assigned to the C-O stretching vibrations of the uronic acids. Particularly, PHPS-D showed another obvious absorption peak at 1730 cm^−1^, which could be explained by the characteristic of the C=O stretching vibration, suggesting the existence of O-acetyl groups contained in PHPS-D [23]. Normally, 1200–800 cm^−1^ were considered as the fingerprint area for carbohydrates. Among these, three characteristic bands at 1101 cm^−1^, 1041 cm^−1^, and 980 cm^−1^ reflected the characteristic property of a pyranose ring in the polysaccharide structure. Two absorption peaks at 883 cm^−1^ and 847 cm^−1^ were observed in the spectra, indicating that both PHPS-D and PHPS-W contained β-glycosidic bonds and α-glycosidic bonds [24], respectively. In addition, the small peak at 810 cm^−1^, belonging to mannose, agreed with the HPLC analysis. The FTIR spectra further confirmed the selectivity of DES, which was prone to extract the specific polysaccharide with more uronic acid.

#### 2.5.5. AFM of Two Prepared Polysaccharides

AFM has been proven effective for exploring the advanced conformation of biomacromolecules at a nanometric level. In this study, the molecular topography of PHPS-D and PHPS-W prepared in 10 μg/mL were observed by AFM, as shown in Figure 7. Through XEI analysis, PHPS-W presented a cloud-like conformation in an aqueous solution, with the height of 7.5–8.5 nm, while PHPS-D appeared as a worm-like chain, with lower height of 1.0–1.6 nm, which was in accordance with the Rg diameters detected from SEC-MALLS. The longer molecular chain reflected that the Chcl-SA system avoided acidic biomacromolecular chain scission [18]. These differences in conformation might be ascribed to the discrepancy between uronic acids contained in the molecular chains [20]. More uronic acids in the structure of PHPS-D would prevent intermolecular chains from aggregating, and cause them to easily disperse in an aqueous due to the existence of ion electric charges. However, single PHPS-Ws may form stacked architecture via intra/inter hydrogen bonds with each other. This phenomenon has been reported in other polysaccharides [25,26]. The observation by AFM elaborated that the carboxylic acid-based DES can promote the release of acidic polysaccharides by the formation of hydrogen bonds via the carboxyl groups in the molecules.

### 2.6. SEM of Raw Material by Different Treatments

Plant tissue can be destroyed, to some extent, by different solvents, which would enable polysaccharides to be released out of the cell wall from the inner matrices [16]. In order to evaluate the performance of DES on the extraction of PHPS from *P. hepiali*, the external microscopic morphology of the raw mycelia powder was observed by SEM before/after two kinds of treatments. From Figure 8A, we can easily see the intact texture of the cell surface wall of the powdered mycelia, with no trace of damage before any treatments. However, the outer powder of *P. hepiali* tended to be damaged, to a certain degree, after two different extractions. The hot-water treatment produced some ridges and wrinkles on the mycelian surface, as shown in Figure 8B, which appeared to be rougher than those in the original materials. Moreover, the tissue matrix presented nearly collapsed after DES treatment, which could induce the mechanical damage on the cell surface. Some pores could be observed to be loosely arranged, as seen in Figure 8C. Those observations confirmed the selective solubilization of polysaccharides by the Chcl-SA system during the preparation, Similar phenomenon have been reported in cellulose extraction and lignocellulosic biomass delignification [27]. The SEM results, elaborated in the above measurements, showed that the rupture degree of the mycelia cells was consistent with the extraction yields of PHPS, and the DES method showed better performance [19].

### 2.7. Immunomodulatory Activity of PHPS on RAW 264.7 Cells

#### 2.7.1. Cell Viability

The cytotoxicity of the PHPS samples on RAW 264.7 cells was firstly investigated by CCK-8 assay in order to elaborate the mechanism of immune activity. As shown in Figure 9A, after the 24 h treatment, the cell viability rates of RAW 264.7 cells in the tested conation (25, 50, 100, 200, 400 μg/mL) were 100.23% and 103.1%, 101.4% and 106.5%, 105.5% and 109.2%, 109.1% and 115.3%, and 112.5% and 116.7% for PHPD-W and PHPS-D, respectively (*p* < 0.05). Our data indicated that neither PHPS-D nor PHPS-W showed toxicity on the RAW 264.7 cells within the tested concentration.

#### 2.7.2. Pinocytotic Activity

Neutral red dye, which can react with lysosome inside the cells to produce red substances, has been widely used as an acid–base indicator for living cells. Once the macrophage cells are activated, the consumption of neutral red dye can be an effective way to assess their pinocytic and phagocytic activity [28]. In comparison with that of the untreated cells, the absorbance intensities of the cells treated with PHPS-D and PHPS-W in the concentration of 25–200 μg/mL were remarkably increased in a dose-dependent manner (Figure 9B). When the concentration was 100 μg/mL, the cell uptake intensity of PHPS-D treated cells hit a value of 165.2%, which was 1.2-fold higher than that of PHPS-W treated cells and 91.6% that of the LPS treated cells (*p* < 0.05). The results indicated that both PHPS-D and PHPS-W showed potent pinocytic activity for the RAW264.7 cells, and that the former could exhibit better performance. Considering the structural differences between the two polysaccharides, the reason might be that more extended molecular chains of PHPS-D prepared by the Chcl-SA system could more easily bind to the related receptors on the cell surface, resulting in subsequent signal transduction and the initiation of pinocytotic function [29].

#### 2.7.3. Production of NO, IL-6, TNF-α, and IL-1β Cytokines

The activated macrophage could play a role in immunomodulatory function by secreting a variety of inflammatory cytokines, including NO, IL-6, TNF-α, IL-1β, etc. NO is one of the important signaling transduction mediums, which is involved in the host defense in the form of lethality, apoptosis, and neurotransmission. Other cytokines are types of bioactive key proteins produced by the stimulation of either antigen or mitogen in RAW 264.7 cells. They normally participate the process of cell proliferation, differentiation, and relative receptor expression [29]. 

Therefore, the effects of different prepared PHPS samples on the secretion of those cytokine representatives were measured in this study. It was easy to observe that both PHPS-D and PHPS-W in the range of 25–200 μg/mL significantly promoted the NO, IL-6, TNF-α, and IL-1β secretion levels of RAW264.7 cells in a dose-dependent manner, as shown in Figure 10. At the concentration of 100 μg/mL, the concentration of NO, IL-6, TNF-α, and IL-1β after treatment with PHPS-D and PHPS-W were measured to be 27 μmol and 18 μmol, 4438 pg/mL and 3216 pg/mL, 1129 pg/mL and 877 pg/mL, and 47 pg/mL and 36 pg/mL, respectively. More specifically, the secretion of all the cytokines in the RAW 264.7 cells exposed to PHPS-D and PHPS-W at 200 μg/mL were comparable to those of LPS at 10 μg/mL (*p* < 0.01), especially PHPS-D. Considering the differences in the structural properties between PHPS-D and PHPS-W, the stronger immunostimulatory activity might be due to the higher uronic acid contents and lower molecular weight inside the biopolymer prepared by DES extraction. Similar findings have been reported in other studies [19]. All the data showed that PHPS-D could better stimulate the secretion of pro-inflammatory cytokines in RAW264.7 than could PHPS-W. With respect to more carboxyl groups in the chain of PHPS-D, the tailorable DES might boost the molecular recognition of PHPS-D in the immune system, consistent with other acidic polysaccharides [18,28].

#### 2.7.4. RT-qPCR Analysis

It is well known that some key immune-related genes mediate the macrophage regulation after drug treatment. In order to clarify the mechanism of PHPS samples to trigger immune cells, the gene expression of mRNA of iNOS, IL-6, TNF-α, and IL-1β were measured by RT-qPCR assay. As presented in Figure 11, a significant increase in the mRNA expression levels of iNOS and the other three cytokines was observed in the activated macrophage cells for both PHPS-D and PHPS-W, comparable to that of the negative control. The data agreed well with the secretion levels of NO, IL-6, TNF-α, and IL-1β. Moreover, the mRNA expression values of iNOS, IL-6, TNF-α, and IL-1β exposed to PHPS-D at 200 μg/mL were 1.12, 1.16, 1.18, and 1.2 times those of PHPS-W, respectively. The results were consistent with our previous findings [30,31,32] and further confirmed that both PHPS-D and PHPS-W exercised their immunomodulatory activities by upregulations of mRNA of iNOS and three other proinflammatory cytokines, particuarly the former.

## 3. Materials and Methods

### 3.1. Materials and Reagents

The fermented mycelia of *P. hepiali* was provided by Zhejiang Bioasia Life Science Institute; glucose, 95% alcohol, choline chloride (ChCl), 1,2-propylene glycol, glycerol, D-sorbitol, citric acid, succinic acid, urea, and other reagents were obtained from Sinopharm Chemical Reagent Co., Ltd. (Shanghai, China). Mannose(Man), Fucose (Fuc), Glucuronic acid (GluUA), Galacturonic acid(GalUA), Rhamnose(Rham), Ribose(Rib), Galactose (Gal), Xylose (Xyl), Glucose (Glu) and Arabinose (Ara) were purchased from Shanghai Aladdin Bio-Chem Technology Co., Ltd. (Shanghai, China). Lipopolysaccharide, polymyxin B (PMB), IL-6, IL-1β, and TNF-α ELISA kits were purchased from HuaMei company (Wuhan, China). All other chemicals were at analytical grade.

### 3.2. Preparation of DESs

The DESs were prepared by mixing two HBAs (choline chloride and betaine) with various HBDs, including polyol, organic acid, and acid amide, following the procedure shown in Table 4. Subsequently, the obtained mixture was stirred and heated in an oil bath at 80 °C with a magnetic agitator until a transparent and uniform liquid formed [16]. After cooling down, the liquid was sealed for the following experiments.

### 3.3. Extraction of PHPS by DESs or Traditional Methods

The fermented mycelia of *P. hepiali* was dried in an oven at 60 °C for 6 h until the moisture content was below 10%. Then, the dried mycelia were crushed by a high-speed multifunctional grinder to produce the crude P. batatas powder, which was put through a 100-mesh sieve. A total of 5.000 g of *P. hepiali* powder was accurately weighed in a conical flas, and extracted with different solvents (water or DESs) under the designed DES conditions, or a 90 °C water bath for 2 h. Next, the mixture was deproteinated using the Sevag method [33]. After centrifugation, four times of anhydrous ethanol was added into the solution (80% final concentration) and precipitated for at least 12 h. The precipitate was obtained by centrifugation and redissolved in distilled water for further purification through the membrane dialysis method (MWCO of 8000 Da, distilled water flushing for 48 h). Then, the final extracted PHPS were obtained by lyophilization.

### 3.4. Selection of Molar Ratio and Water Content in DESs

In order to obtain the ideal DES system for maximum PHPS extraction, the molar ratio and water content of the selected DES were studied prior to the optimization. Under the other fixed conditions, different molar ratios (1:1, 1:2, 1:3, 1:4, 1:5, and 1:6) of solvent and a series of water contents (0%, 10%, 20%, 30%, 40%, and 50%) were tested for the acquisition of the highest PHPS yields. 

### 3.5. Chemical Composition of PHPS

The polysaccharide contents of the samples were determined by the phenol-sulfuric acid method using a glucose standard curve. The standard curve was drawn using glucose as the standard to yield a regression equation (y = 6.9975 x+ 0.0073, R^2^ = 0.9996) [34]. The content of uronic acid was tested by the m-hydroxybiphenyl method using D-galacturonic acid as the standard [35]. The protein content was measured by BCA assay. The extraction yield of PHPS was calculated according to Formula (2): (2)$$\mathrm{Extraction}\;\mathrm{yield}\;(\%)=\frac{c\times V\times N}{m}\;\times \;100\%$$
where *c* is the concentration of PHPS from above standard curve (mg/mL), *V* is the sample volume (mL), *N* is the dilution multiple, and *m* is the weight of the crude powder (g).

### 3.6. Single Factor Design

After the establishment of the DES component, three different single conditions were investigated, which were liquid–solid ratio (10:1 mL/g, 20:1 mL/g, 30:1 mL/g, 40:1 mL/g, 50:1 mL/g, and 60:1 mL/g), extraction temperature (50 °C, 60 °C, 70 °C, 80 °C, 90 °C, and 100 °C), and extraction time (0.5 h, 1.0 h, 1.5 h, 2.0 h, 2.5 h, and 3.0 h). Th average value of triplicate experiments was chosen for the proper determination of each experimental study.

### 3.7. Optimization of Extraction Process

Based on the results of the single factor investigation, a 3-factor, 3-level Box–Behnken design (BBD), with 5 center points, was employed to further study the effects of different variables and their interactions on the PHPS-D yield. The basic information for three independent factors was liquid–solid ratio (X_1_), extraction temperature (X_2_), and extraction time (X_3_), with the detailed formula shown in Table 5. Response surface methodology (RSM) was performed to optimize the extraction process of PHPS-D by using Design Expert 12 software (State-Ease, Minneapolis, MN, USA) [14,15]. 

### 3.8. Physicochemical Properties

#### 3.8.1. Molecular Weight Determination

A total of 3 mg/mL of polysaccharide was dissolved in Milli-Q water and subjected to separation by a multi-angle laser scattering system (MALLS) (DAWN HELEOS-II, Wyatt Technology Corporation, Santa Barbara, CA, USA), coupled with a TSK gel G3000PWXL column (7.5 × 600 mm, TOSOH Corporation, Japan) and a differential refractive detector (RI-10A, Shimazu Corporation, Kyoto, Japan), at 25 °C [31]. The mobile phase was a 0.15 mol/L NaNO_3_ solution containing 0.02% NaN_3_, with a flow rate of 0.6 mL/min. The value of 0.1380 mL/g was adopted as the refractive index increment dn/dc. ASTRA 5.3.4 software was used for the calculation of Mw and the radius of gyration (Rg).

#### 3.8.2. Monosaccharide Composition

The chemical composition of monosaccharides was measured by HPLC after the preparation of PMP derivatives following the procedure in our previous studies [36]. A total of 5 mg of polysaccharides were hydrolyzed by 4 mol/L trifluoroacetic acid (TFA) at 121 °C for 5 h. After the removal of TFA residues by methanol and Milli-Q water, the hydrolyzed solution of the samples was ready for PMP derivation. The reaction of 10 monosaccharide standards or hydrolyzed samples was started with the addition of 150 μL of 0.6 mol/L NaOH solution and 300 μL of 0.5 mol/L 1-phenyl-3-methyl-5-pyrazolone (PMP) methanol solution at 70 °C for 2 h. Next, 300 μL of 0.3 mol/L HCl was added to neutralize the solution, followed by the removal of the PMP residue with 3 folds of chloroform(*v/v*). The HPLC system with an Eclipse XDB-C18 column (4.6 × 250 mm × 5 μm, Agilent, Santa Clara County, CA, USA) was used to analyze the liquid. The mobile phase was 83:17 (*v/v*) mixture of 0.05 mol/L PBS (pH 6.9) and acetonitrile. A total of 1.0 mL/min was operated as the flow rate, and the detection wavelength was set at 245 nm.

#### 3.8.3. FT-IR Analysis

A 13 mm pellet was prepared for FT-IR testing, with a mixed powder of polysaccharide sample and KBr (2 mg/150 mg), which was scanned immediately by a Nicolet iS50 FT-IR spectrometer (Thermofishier Corporation, Fremont, CA, USA). The data were collected in the range of 4000–400 cm^−1^, with 32 times scanning at a 4 cm^−1^ resolution.

#### 3.8.4. Conformation by AFM

PHPS-D and PHPS-W were thoroughly dissolved in Milli-Q water at a stock solution of 1 mg/mL with 2 h stirring. Then, they were diluted stepwise into 10 μg/mL with Milli-Q water. After 0.22 μm membrane filtration, 2 μL of sample was dropped onto freshly cleaved mica and air-dried at room temperature. The topography was tested by AFM (XE-70, Park Scientific Instruments, Suwon, Korea) in a tapping mode. The image was processed by XEI 4.5 software [31].

### 3.9. Microstructure of SEM

The morphology of differently treated raw resources was observed by SEM (S-3400N, HITACHI, Kyoto, Japan). Each photograph was scanned with 500× magnification under the voltage of 12.5 Kv [16].

### 3.10. Immunomodulatory Activity

#### 3.10.1. Cell Viability and Cytotoxicity of RAW264.7

The mouse mononuclear macrophage RAW264.7 cell line was cultured in DMEM containing 10% FBS, 1% penicillin-streptomycin, and 20 μg/mL polymyxin B (PMB) in a humidified atmosphere at 37 °C with 5% CO_2_. A CCK-8 kit was used to measure the cell viability of RAW264.7 macrophages [37]. After 24 h incubation in 96-well plates, the cell culture supernatants were carefully removed and treated with 6.25–200 μg/mL PHPS samples or positive LPS (10 μg/mL) for 24 h. A total of 20 μg/mL of PMB was used to avoid LPS contamination. Next, CCK-8 solution (10 μL) was added to the system for 1 h of maintenance at 37 °C. The 450 nm of absorbance was determined immediately using a Thermofishier Scientific Microplate Reader.

#### 3.10.2. Pinocytic Assay

The neutral red assay was used to evaluate the effects of the PHPS samples on the pinocytic activity of the RAW264.7 cells [38]. Initially, macrophage cells with a density of 1 × 10^6^ cells/mL were incubated with DMEM medium for 24 h. Then, 25–200 μg/mL of two different PHPS samples were loaded for another 24 h. At the same time, 10 μg/mL of LPS was labeled as the positive group. Afterwards, 100 μL of 0.1% neutral red was added for 1 h maintenance, followed by PBS washing three times, a 100 μL solution of proportional glacial acetic acid and ethanol was selected for cell lysis. Finally, the absorbance at 540 nm was determined after statical reaction for 2 h.

#### 3.10.3. Cytokine Production of PHPS

The production of the NO, IL-6, TNF-α, and IL-1β cytokines was measured according to the methods used in our previous methods [32]. The macrophages were exposed to PHPS samples with 25–200 μg/mL or LPS (10 μg/mL) for 24 h incubation. Then, the supernatants were harvested for the next analysis. The Griess method was used for the determination of macrophage NO content. The levels of IL-6, TNF-α, and IL-1β were judged with the help of commercial ELISA kits, following the manufacturer’s instructions. 

#### 3.10.4. RT-PCR Analysis

The gene expression of the activated immune cells was tested using the RT-qPCR methods mentioned in the previous study [37]. The macrophages were incubated in 6-wells plates at 37 °C, followed by the treatment with different concentrations of PHPS samples (25–200 μg/mL) or LPS (10 μg/mL). Then, mRNA analysis was conducted using the commercial protocol with a series of processes, including cell collection, RNA extraction, cDNA reverse transcription, and DNA amplification. The cDNA encoding iNOS, IL-6, TNF-α, and IL-1β were amplified by using GOTaq Flexi DNA polymerase and specific primers, according to the previously reported sequences. GAPDH was selected as the internal referent, and the 2^−∆∆Ct^ method was used to calculate the mRNA expression levels. The final gene amplification was quantitatively measured at ABI 7500 using a real-time PCR instrument.

### 3.11. Statistical Analysis

The data were expressed as mean ± standard error. Differences were tested by ANOVA. Differences with *p* < 0.05 were considered significant, and differences with *p* < 0.01 were considered extremely significant.

## 4. Conclusions

In this study, a green and tailorable DES system consisting of Chcl-SA in the molar ratio of 1:3, with a 30% water content, was employed for the extraction of acidic polysaccharides (PHPS-D) from raw mycelia of *P. hepiali*. The optimal condition was presented as follows: liquid–solid ratio of 50 mL/g, extraction temperature of 85 °C, and extraction time of 1.7 h. Under these conditions, the actual PHPS-D yield was 12.78 ± 0.17%, which was significantly higher than that of PHPS-W. The physicochemical properties indicated that there was more uronic acid (22.34 ± 1.38%) and glucose (40.3 ± 0.5%) contained in PHPS-D than in PHPS-W, as did the higher molecular weight (3.26 × 10^5^ g/mol) and longer Rg (78.2 ± 3.6 nm). The AFM data reflected that the selectively prepared PHPS-D appeared to be more unfolded and flexible, as a worm-like chain, in the saline solution than the traditionally treated PHPS-W. The results of immunomodulation proved that this tailored PHPS-D could better enhance the secretion levels of NO, IL-6, TNF-α, and IL-1β via activating the corresponding mRNA expression in the RAW264.7 macrophage cells than could PHPS-W. These findings provided a new strategy for preparing high-valued biomacromolecules with a selective DES system, helping us to fully explore the treasured mycelia for use in healthy foods using a green and environmentally friendly approach.

## Figures and Tables

**Figure 1 molecules-27-08020-f001:**
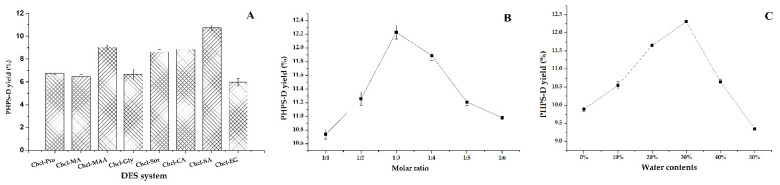
Effects of different DESs on the extraction of PHPS-D yield: (**A**) effect of the DESs systems on PHPS-D yield; (**B**) effect of molar ratio on PHPS-D yield; (**C**) effect of water contents on PHPS-D yield.

**Figure 2 molecules-27-08020-f002:**
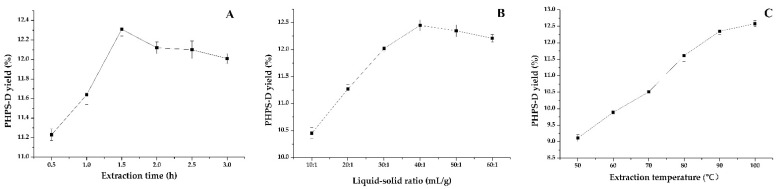
Effect of various single factors on the extraction yield of PHPS-D: (**A**) effect of extraction time on PHPS-D yield; (**B**) effect of liquid to solid ratio on PHPS-D yield; (**C**) effect of extraction temperature on PHPS-D yield.

**Figure 3 molecules-27-08020-f003:**
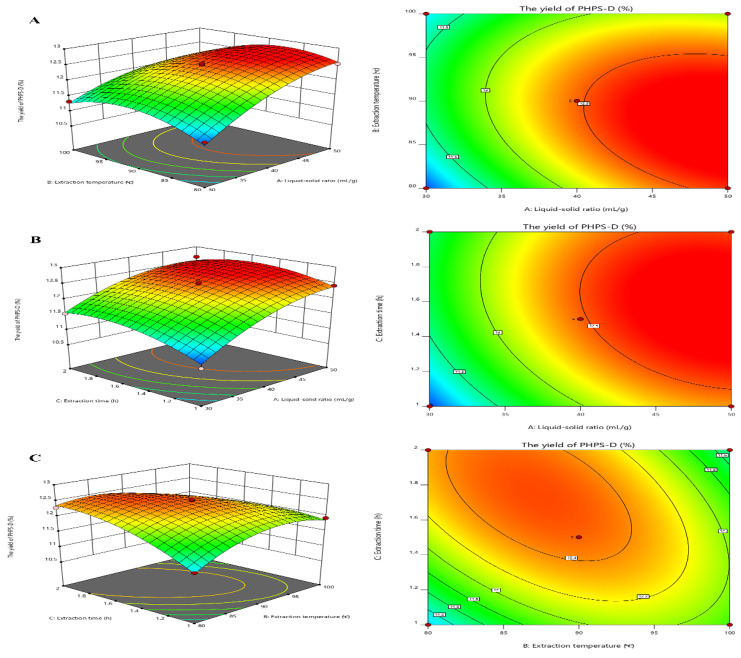
Plots of three-dimensional response surface and two-dimensional contour of the tailorable Chcl-SA-assisted extraction for PHPS-D yield: the interactive effects of liquid to solid ratio and extraction temperature (**A**); the interactive effects of liquid to solid ratio and extraction time (**B**); and the interactive effects of extraction temperature and time (**C**).

**Figure 4 molecules-27-08020-f004:**
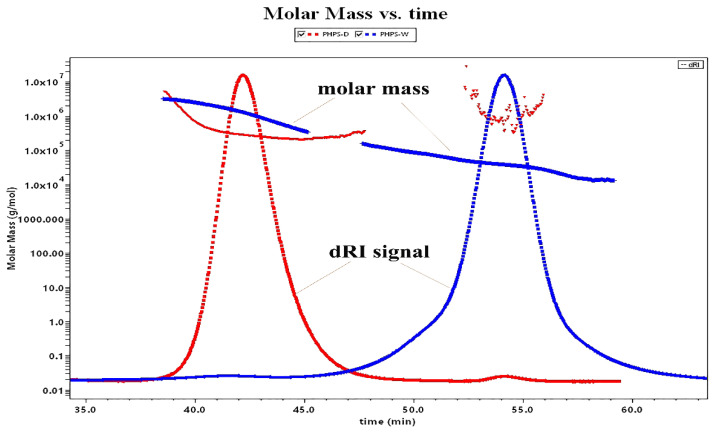
Molar mass profiles of the SEC-MALLS chromatogram for PHPS-D and PHPS-W in 0.15 mol/L NaNO_3_ at 25 °C.

**Figure 5 molecules-27-08020-f005:**
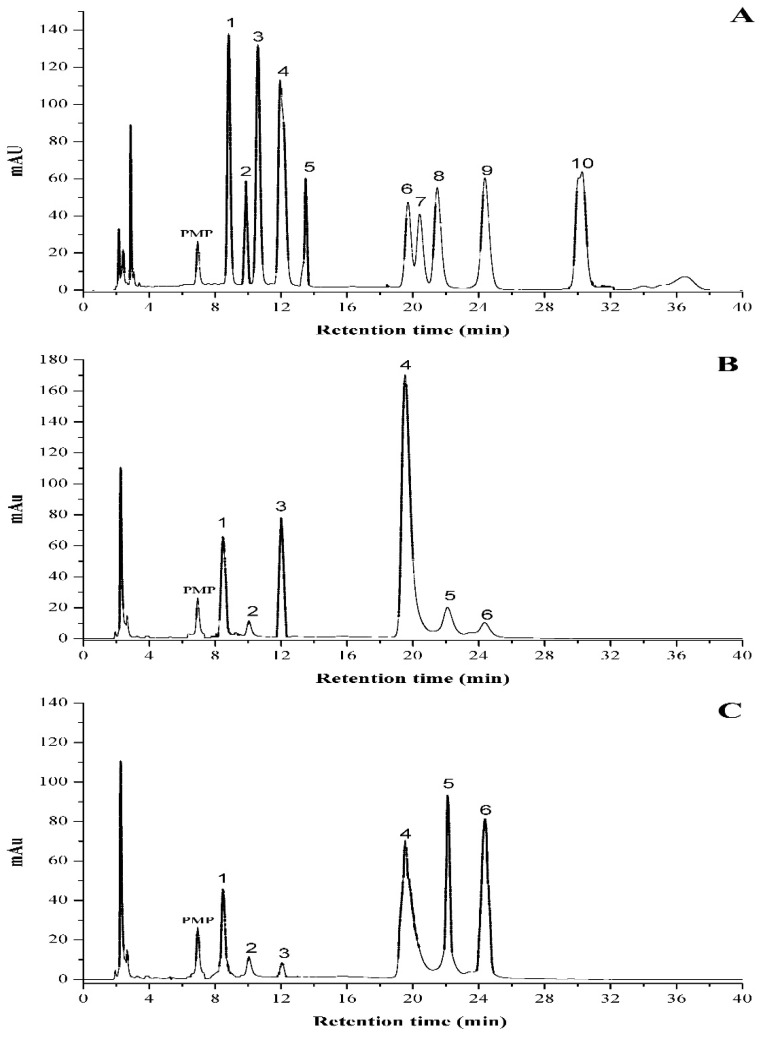
HPLC profiles of PMP derivatives of standard monosaccharides (**A**) PHPS-D and (**B**) PHPS-W. (**C**) Peaks: 1. mannose; 2. ribose; 3. rhamnose; 4. glucuronic acid; 5. galacturonic acid; 6. glucose; 7. xylose; 8. galactose; 9. arabinose; 10. fucose.

**Figure 6 molecules-27-08020-f006:**
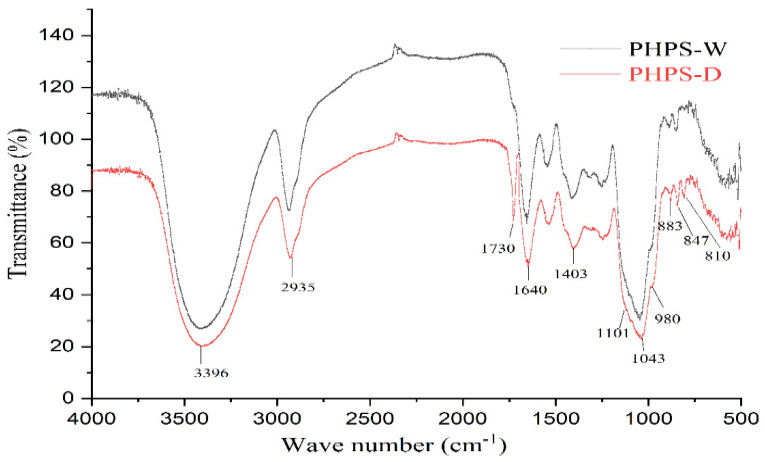
FT-IR spectra of PHPS-D and PHPS-W detected in the range of 4000–400 cm cm^−1^ at 4 cm cm^−1^ resolution.

**Figure 7 molecules-27-08020-f007:**
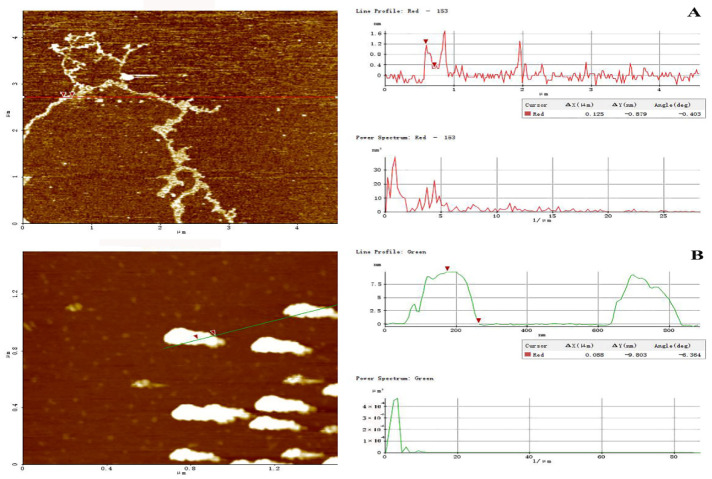
Tapping-mode AFM images of PHPS-D (**A**) and PHPS-W (**B**) at 10 μg/mL on mica, with scan size of 5 μm × 5 μm. The cross-sectional profile is shown on the right side of each AFM image, and the curves with different colors in the cross-sectional profile correspond to the line of the same color in the AFM images on the left.

**Figure 8 molecules-27-08020-f008:**
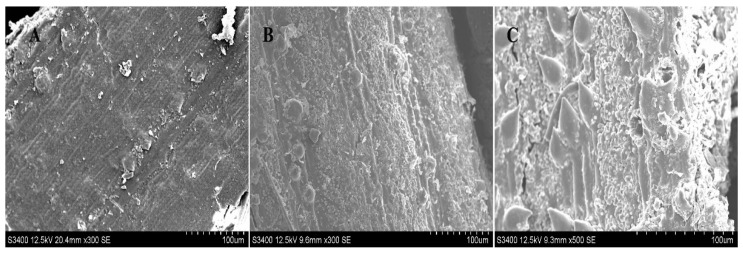
SEM of raw *Paecilomyces hepiali* powder before treatment (**A**), and after hot water extraction (**B**), and DES extraction (**C**), with magnification at 500× at 12.5 kv.

**Figure 9 molecules-27-08020-f009:**
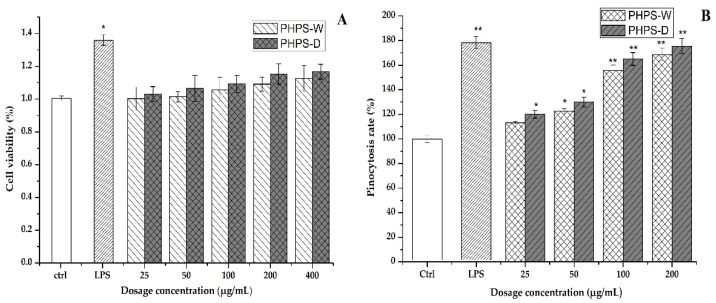
Effects of PHPS-D and PHPS-W treatments on RAW264.7 cells: cell viability (**A**), and pinocytosis rate of neutral red (**B**). Note: Each experiment was tested in triplicate, and the error bars are standard deviations, Significant differences in cell viability and pinocytosis rates of the samples vs. the control group are presented by * *p* < 0.05 and ** *p* < 0.01.

**Figure 10 molecules-27-08020-f010:**
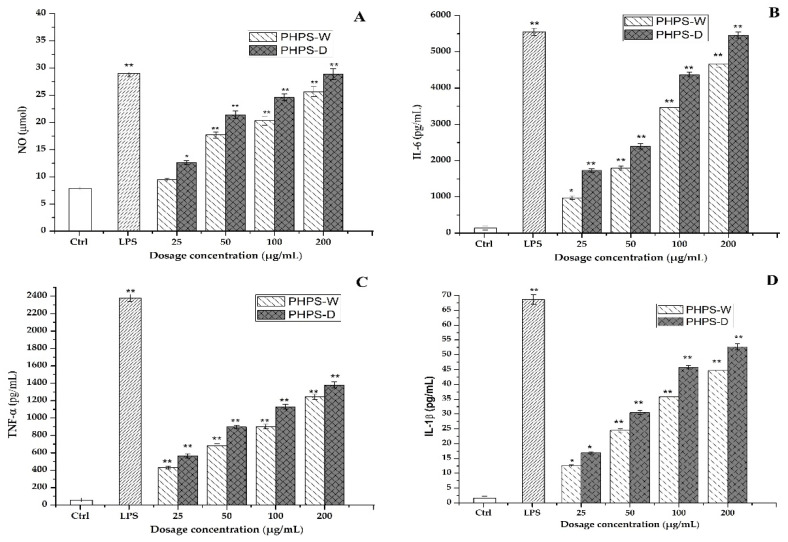
Effect of PHPS-D and PHPS-W treatments on the secretion levels in the RAW264.7 cells of NO (**A**), IL-6 (**B**), TNF-α (**C**), and IL-1β (**D**). Note: Each experiment was tested in triplicate, and the error bars are standard deviations, Significant differences regarding cytokine release in the samples vs. control group are represented by * *p* < 0.05 and ** *p* < 0.01.

**Figure 11 molecules-27-08020-f011:**
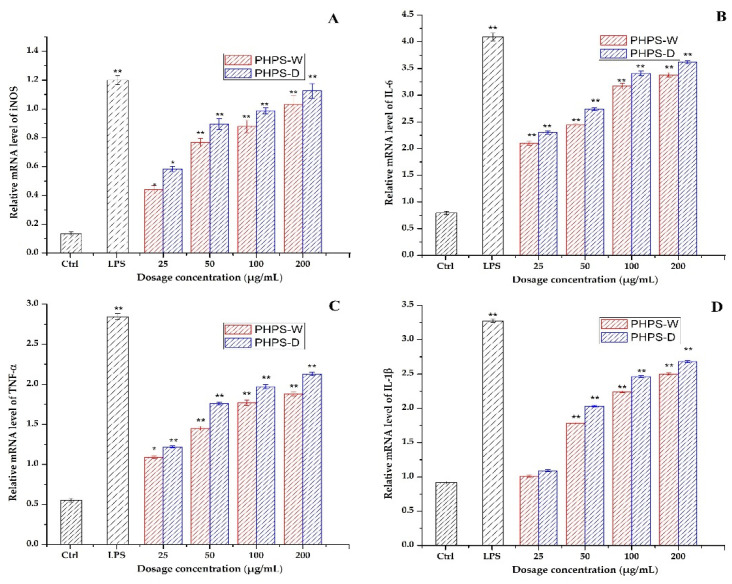
Effect of PHPS-D and PHPS-W treatments on RAW264.7 cells mRNA expression levels of NO (**A**), IL-6 (**B**), TNF-α (**C**), and IL-1β (**D**). Note: Each experiment was tested in triplicate, and the error bars are standard deviations, Significant differences in the relative mRNA levels in the samples vs. the control group are represented by * *p* < 0.05 and ** *p* < 0.01.

**Table 1 molecules-27-08020-t001:** Design and results of the response surface method.

Run	Variable Levels			Extraction Yield of PHPS-D/%	
	X_1_ (Liquid–Solid Ratio/mL/g)	X_2_ (Extraction Temperature/°C)	X_3_ (Extraction Time/h)	Experimental	Predicted
1	30	80	1.5	11.07	10.97
2	50	80	1.5	12.51	12.57
3	30	100	1.5	11.32	11.27
4	50	100	1.5	11.95	12.05
5	30	90	1.0	10.84	10.96
6	50	90	1.0	12.41	12.38
7	30	90	2.0	11.54	11.57
8	50	90	2.0	12.65	12.53
9	40	80	1.0	11.23	11.21
10	40	100	1.0	11.92	11.85
11	40	80	2.0	12.28	12.35
12	40	100	2.0	11.45	11.47
13	40	90	1.5	12.52	12.47
14	40	90	1.5	12.46	12.47
15	40	90	1.5	12.36	12.47
16	40	90	1.5	12.53	12.47
17	40	90	1.5	12.49	12.47

**Table 2 molecules-27-08020-t002:** Variance analysis results of the regression model.

Source	Sum of Squares	df	Mean Square	F-Value	*p*-Value
Model	5.75	9	0.64	52.63	<0.0001 **
X_1_	2.82	1	2.82	232.25	<0.0001 **
X_2_	0.025	1	0.025	2.08	0.1920
X_3_	0.29	1	0.29	23.78	0.0018 **
X_1_X_2_	0.16	1	0.16	13.51	0.0079 **
X_1_X_3_	0.053	1	0.053	4.36	0.0753
X_2_X_3_	0.58	1	0.58	47.56	0.0002 **
X_1_^2^	0.40	1	0.40	33.27	0.0007 **
X_2_^2^	0.85	1	0.85	70.13	<0.0001 **
X_3_^2^	0.38	1	0.38	31.68	0.0008 **
Residual	0.085	7	0.012		
Lack of Fit	0.066	3	0.022	4.73	0.0836
Pure Error	0.019	4	4.670 × 10^−3^		
Cor Total	5.84	16			

** means the significance *p* < 0.01.

**Table 3 molecules-27-08020-t003:** Physicochemical properties of different polysaccharides prepared from *Paecilomyces hepiali* powder.

	PHPS-D ^a^	PHPS-W ^a^
Chemical compositions		
Extraction yield (%)	12.78 ± 0.17	8.67 ± 0.21
Total sugars (%)	80.92 ± 1.23	74.53 ± 2.36
Protein (%)	4.49 ± 0.74	6.75 ± 0.97
Uronic acid (%)	22.34 ± 1.38	5.28 ± 0.16
Molecular characteristics		
Mw × 10^4^ (Da) ^b^	32.6 ± 2.4	4.37 ± 1.3
Rg (nm) ^c^	78.2 ± 3.6	20.1 ± 2.7
Monosaccharides components		
Mannose	16.7 ± 0.5	14.2 ± 0.6
Ribose	4.1 ± 0.9	7.8 ± 0.7
Glucuronic acid	22.5 ± 0.4	4.9 ± 0.8
Glucose	40.3 ± 0.5	30.1 ± 0.4
Galactose	10.2 ± 0.2	20.6 ± 0.5
Arabinose	6.2 ± 0.3	22.4 ± 0.7

^a^ PHPS-D and PHPS-W were the polysaccharides prepared by DES and hot water; ^b^ absolute molecular weight (Mw); ^c^ radium of gyration (Rg).

**Table 4 molecules-27-08020-t004:** List of preparation of DESs employed in this study.

NO.	HBA	HBD	Mole Ratio	Water Content	Abbreviation
DES-1	choline chloride	1,2-propylene glycol	1:1	10%	Chcl-Pro
DES-2	choline chloride	malonic acid	1:1	10%	Chcl-MA
DES-3	choline chloride	DL-malic acid	1:1	10%	Chcl-MAA
DES-4	choline chloride	glycerol	1:1	10%	Chcl-Gly
DES-5	choline chloride	D-sorbitol	1:1	10%	Chcl-Sor
DES-6	choline chloride	citric acid	1:1	10%	Chcl-CA
DES-7	choline chloride	succinic acid	1:2	10%	Chcl-SA
DES-8	choline chloride	urea	1:2	10%	Chcl-UR
DES-9	choline chloride	1,4-butanediol	1:1	10%	Chcl-Bu4
DES-10	choline chloride	ethylene glycol	1:1	10%	Chcl-EG
DES-11	choline chloride	1,3-butanediol	1:2	10%	Chcl-Bu3
DES-12	betaine	1,2-propylene glycol	1:2	10%	Bet-PG
DES-13	betaine	1,3-butanediol	1:2	10%	Bet-Bu3
DES-14	betaine	urea	1:2	10%	Bet-UR
DES-15	betaine	malonic acid	1:2	10%	Bet-MA

**Table 5 molecules-27-08020-t005:** Factors and levels for the Box–Behnken response surface method.

Variable	Units	Coded Levels			
		Symbol	−1	0	1
Liquid–solid ratio	mL/g	X_1_	30	40	50
Extraction temperature	°C	X_2_	80	90	100
Extraction time	h	X_3_	1.0	1.5	2.0

## Data Availability

Data are contained within the article.

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
