# Peer review of "Natural Deep Eutectic Solvent-Assisted Extraction, Structural Characterization, and Immunomodulatory Activity of Polysaccharides from Paecilomyces hepiali"

_molecules, 2022, doi:10.3390/molecules27228020_

Round 1

Reviewer 1 Report

In this study, green treatment was described and optimized for the extraction of polysaccharides from raw mycelia of Paecilomyces hepialid, and compared to standard one with hot water extraction. The described method gives better polysaccharide yield, physicochemical properties, and particular immunomodulation assay results. Some major remarks should be addressed:

The scientific explanation for the behavior obtained missing, which is highly important for publishing in the prestige journal Molecules. Please provide a possible scientific explanation and underline it throughout the manuscript, without adequate explanation I will not suggest publication.

All pictures have poor resolution and figure capture should be expanded.

Define Rg in the Abstract and where it is used first.

Explain Figure 4 and additional lines (besides two peaks with maxima). Please clarify the picture.

The results obtained by chromatography are not coordinated. What about 4 (glucuronic acid), 5, 6..? Explain obtained shift in retention time. The numbers do not correspond to standard monosaccharides…

Provide an explanation of the right side of AFM Figure 7. The Y-axis is unreadable.

Author Response

Response to the Reviewers’ Comments

Dear Editor,

Thank you very much for your letter and the reviewers’ reports. Based on your comment and request, we have carefully proof-read and made extensive modification on the original manuscript.

List of Actions:

1: The revised paragraphs are in red type.

2: The new figure and some supplements are revised in red type.

3: Response to editor’s comments are appended below.

Here, we attach revised manuscript for your approval. A document answering every question from the editor is also summarized and enclosed, and our description on revision according to the reviewers’ comments is appended below. We appreciate for editor’s warm work earnestly, and hope that the corrections will meet with approval.

Looking forward to hearing from you.

Sincerely yours,

Liang He

*****************************************************

Reviewer 1: In this study, green treatment was described and optimized for the extraction of polysaccharides from raw mycelia of Paecilomyces hepiali, and compared to standard one with hot water extraction. The described method gives better polysaccharide yield, physicochemical properties, and particular immunomodulation assay results. Some major remarks should be addressed:

Question 1: The scientific explanation for the behavior obtained missing, which is highly important for publishing in the prestige journal Molecules. Please provide a possible scientific explanation and underline it throughout the manuscript, without adequate explanation I will not suggest publication.

Comment: Thank you so much for your suggestion. As a matter of fact, there are some valuable scientific findings in our work, which have been observed and elaborated in the designed experiments. On the one hand, deep eutectic solvent (DES) has attracted great interests on the green technology for the purpose to improve valorization of raw materials and reduce negative influence of human involvement. It exhibits low costs, biopolymer dissolution ability, biodegradability, non-toxicity, and recyclability. Those properties can be ascribed to large asymmetric ions of HBA and low lattice energy of HBD. The character of large size of cations and their conformational flexibility empower itself to have tailorable physicochemical properties, which can affect reaction selectivity, avoid biopolymer chain scission and highly dissolve target. On the other hand, the biological activities of polysaccharides are closely correlated with their structural information including molecular weight, monosaccharide components, flexible branches and glycosidic linkages. The selected system (Chcl-carboxylic acid based DES) in our study has successfully extracted one acidic polysaccharide (PHPS-D) from mycelia of Paecilomyces hepiali, which has larger molecular weight, more uronic acid content, more extended chain conformation than those of commonly obtained polysaccharide (PHPS-W). Moreover, PHPS-D showed better immunomodulatory activity due to this unique structural feature. Our results strongly evidenced that DES could selectively dissolving some special biomacromolecules by unique binding of hydrogen bonds or other chemical forces. It becomes a promising tailorable green solvent in the field of health-food preparation. All this scientific explanation has been underlined in the related parts through the paper.

Question 2: All pictures have poor resolution and figure capture should be expanded.

Comment: Thanks for your suggestion. We have already improved the resolution of all the figures and provided the detailed information in every figure capture.

Question 3: Define Rg in the Abstract and where it is used first.

Comment: Thanks for your valuable suggestion. Rg refers to radium of gyration for single chain of biomacromolecule. We have already replaced Rg with full name where it first appeared in the paper.

Question 4: Explain Figure 4 and additional lines (besides two peaks with maxima). Please clarify the picture.

Comment: Thanks for your valuable suggestion. We have added the detailed information in both Figure 4 and related parts in paper. Figure 4 showed the molar mass profiles of the SEC-MALLS chromatogram for PHPS-D and PHPS-W eluted in 0.15 mol/L NaNO3 at 25℃. Each peak means the RI signals of the concentration of every sample at a certain retention time when it is eluted from the TSK gel G3000PWXL column (7.5×600 mm, TOSOH Corporation, Japan). And the additional relative straight line represents the certain value of molar mass for each polysaccharide at corresponding eluted time and RI signal. It can be found that the red line reflects the molar mass of every eluted PHPS-D particle was in the range of 1.0×105 to 1.0×106 g/mol at the whole elution process. While the blue line indicates the molar mass value of PHPS-W particle drops down into the range of 1.0×104 to 1.0×105 g/mol from 45 min to 60 min although it is similar with that of PHPS-D before 45 min. Consequently, the average molar mass weight of PHPS-D and PHPS-W were 32.6 ± 2.4 and 4.37 ± 1.3 Da in Table 5, respectively.

Question 5: The results obtained by chromatography are not coordinated. What about 4 (glucuronic acid), 5, 6..? Explain obtained shift in retention time. The numbers do not correspond to standard monosaccharides…

Comment: Thanks for your suggestion. We have already re-coordinated the chromatograph in Figure 5. Number 4 means the glucuronic acid, number 5 is the eluted peak of galacturonic acid and number 6 refers to glucose. All the numbers of peaks have been indicated in the figure 5 legend. There was slight shift in some retention time compared with those of the standard sugar. It might be caused by the tiny fluctuation of liquid system pressure. Concerning the good degree of separation of each sugar in the HPLC profile, that drawbacks could be negligible. Now the monosaccharide numbers of PHPS-D and PHPS-W in the new coordinated profile are consistent with the standard sugars.

Question 6: Provide an explanation of the right side of AFM Figure 7. The Y-axis is unreadable.

Comment: Thanks for your suggestion. We have already provided the explanation of the right side of AFM in Figure 7 legend. The cross-sectional profile is shown on the right of each AFM image, and the curves with different colors in the cross-sectional profile correspond to the line of the same color in the AFM images on the left, which means the height of the sample polymer chain on mica.

Once again, special thanks for your comments, and we are looking forward to learning much more from you.

Yours sincerely

Liang He

Reviewer 2 Report

The manuscript “Natural Deep Eutectic Solvent Assisted Extraction, Structural Characterization and Immunomodulatory Activity of Polysaccharides from Paecilomyces hepiali" is devoted to the isolation of polysaccharides from entomophagous fungus using NADES of various compositions. The effectiveness of 15 different NODES formulations was evaluated, the extraction process was studied in detail, and the extraction conditions were optimized in accordance with BBD DoE. The isolated polysaccharides were characterized by various methods. The assessment of biological activity by various methods is very interesting.

The results obtained can be useful from the point of view of biochemistry and physiology.

I think, this manuscript can be published in the Molecules after minor revision taking into account general recommendation and some of the remarks described below:

1.      The inscriptions on the drawings are poorly readable, the font is too small.

2.      3.2. Preparation of DESs: At what stage of the preparation of DESs was water added? Why was the percentage of 10% chosen? DES with citric acid is quite viscous and it is very difficult to work with it during extraction.

3.      It is necessary to explain why the paper provides a combination of Single Factor Evaluation and BBD Analysis? Using Single Factor Evaluation, you can reduce the number of parameters optimized in BBD. However, the set of parameters is duplicated in the work.

Author Response

 Response to the Reviewers’ Comments

Dear Editor,

Thank you very much for your letter and the reviewers’ reports. Based on your comment and request, we have carefully proof-read and made extensive modification on the original manuscript.

List of Actions:

1: The revised paragraphs are in red type.

2: The new figure and some supplements are revised in red type.

3: Response to editor’s comments are appended below.

Here, we attach revised manuscript for your approval. A document answering every question from the editor is also summarized and enclosed, and our description on revision according to the reviewers’ comments is appended below. We appreciate for editor’s warm work earnestly, and hope that the corrections will meet with approval.

Looking forward to hearing from you.

Sincerely yours,

Liang He

*****************************************************

Reviewer 2: The manuscript “Natural Deep Eutectic Solvent Assisted Extraction, Structural Characterization and Immunomodulatory Activity of Polysaccharides from Paecilomyces hepiali" is devoted to the isolation of polysaccharides from entomophagous fungus using NADES of various compositions. The effectiveness of 15 different NODES formulations was evaluated, the extraction process was studied in detail, and the extraction conditions were optimized in accordance with BBD DoE. The isolated polysaccharides were characterized by various methods. The assessment of biological activity by various methods is very interesting. The results obtained can be useful from the point of view of biochemistry and physiology. I think, this manuscript can be published in the Molecules after minor revision taking into account general recommendation and some of the remarks described below:

Question 1: The inscriptions on the drawings are poorly readable, the font is too small.

Comment: Thanks for your suggestion. We have already improved the resolution of all the figures and enlarged the fonts to be readable, and provided the detailed information in every figure capture.

Question 2: 3.2. Preparation of DESs: At what stage of the preparation of DESs was water added? Why was the percentage of 10% chosen? DES with citric acid is quite viscous and it is very difficult to work with it during extraction.

Comment: Thanks for your advice. As a matter of fact, different ratio of HBA and HBD were initially mixed in the reactor. Afterwhile, when the solid tended into liquid phase, a certain amount of water was supposed to add in the flask for continuedly stirring until a uniform liquid formed. 10% of water content are normally suitable for most different formula of DESs although it might work not well with some reagents in system. Otherwise that unsatisfactory designed-DES would be abandoned. We do agree with your findings that the viscosity of DES-citric acid was very high, which resulted in a low extraction yield at relatively high temperature. Some of DESs performed even worse than that of DES-citric acid in our experiments. The selected system consisting of Chcl-SA had relatively better performance on PHPS extraction after the optimization of its molar ratio and water content.

Question 3: It is necessary to explain why the paper provides a combination of Single Factor Evaluation and BBD Analysis? Using Single Factor Evaluation, you can reduce the number of parameters optimized in BBD. However, the set of parameters is duplicated in the work.

Comment: Thanks for your valuable suggestion. Normally the evaluation of the single factors can help designer to find the significant factor and its optimal level range, which would be greatly helpful for the next response surface methodology (RSM) procedure. It can reduce the experimental times and provide a reasonable value range. Then the response surface method would be applied to optimize the extraction process by BBD analysis based on the results of single factor. In this study, three variables were considered as independent for PHPS-D extraction after the fixation of molar ratio and water content for Chcl-SA system. The evaluation of these three factors gave the valuable test level ranges for RSM. With those help, the BBD experiment was conducted to find the optimal conditions for PHPS-D yield. And the ANOVA analysis reflected the fitted model could explain the extraction process very well. From that point, the single factor evaluation is necessary as well as the BBD analysis.

Once again, special thanks for your comments, and we are looking forward to learning much more from you.

Yours sincerely

Liang He

Round 2

Reviewer 1 Report

Thank you for your corrections. I recommend this manuscript for publication.